# IL-6 and SAA—Strong Predictors for the Outcome in COVID-19 CKD Patients

**DOI:** 10.3390/ijms25010311

**Published:** 2023-12-25

**Authors:** Rumen Filev, Mila Lyubomirova, Boris Bogov, Krassimir Kalinov, Julieta Hristova, Dobrin Svinarov, Lionel Rostaing

**Affiliations:** 1Department of Nephrology, Internal Disease Clinic, University Hospital “Saint Anna”, 1750 Sofia, Bulgaria; mljubomirova@yahoo.com (M.L.); bbogov@yahoo.com (B.B.); 2Faculty of Medicine, Medical University Sofia, 1504 Sofia, Bulgaria; julieta_sd@yahoo.com (J.H.); dsvinarov@yahoo.com (D.S.); 3Head Biometrics Group, Comac-Medical Ltd., 1404 Sofia, Bulgaria; krassimir.kalinov@comac-medical.com; 4Department of Clinical Laboratory, University Hospital “Alexandrovska”, 1431 Sofia, Bulgaria; 5Nephrology, Hemodialysis, Apheresis and Kidney Transplantation Department, Grenoble University Hospital, 38043 Grenoble, France; lrostaing@chu-grenoble.fr; 6Internal Disease Department, Grenoble Alpes University, 38043 Grenoble, France

**Keywords:** IL-6, SAA, COVID-19, AKI, CKD

## Abstract

In this prospective study, we assessed biomarkers of inflammation (IL-6 and SAA) from the serum of 120 COVID-19 patients, of whom 70 had chronic kidney disease. All the samples were taken at emergency-department (ED) admission. Our goal was to relate the biomarkers to the results of death and acute kidney injury. All the patients underwent chest computer tomography to estimate the severity score (0–5), which was performed at hospital admission. Finally, biomarkers were also evaluated in a healthy control group and in non-COVID-19-CKD patients. IL-6 and SAA were statistically different between the subgroups, i.e., they were significantly increased in patients with COVID-19. Both of the biomarkers (IL-6 and SAA) were independently associated with mortality, AKI and a higher grade of pathological changes in the lung’s parenchyma. Both high baseline levels of IL-6 and SAA on hospital admission were highly correlated with a later ventilatory requirement and mortality, independent of hospital stay. Mortality was found to be significantly higher when the chest CT severity score was 3–4, compared with a severity score of 0–2 (*p* < 0.0001). Conclusions: at the admission stage, IL-6 and SAA are useful markers for COVID-19 patients with CKD.

## 1. Introduction

Since the outbreak of the COVID-19 pandemic, almost 7 million people have died up until this moment (https://covid19.who.int/?mapFilter=deaths (accessed on 13 December 2023). From a medical point of view, the situation in Bulgaria throughout the pandemic was not too much different from the situation in most of the world, but according to official data, the percentage of people vaccinated in Bulgaria is one of the lowest in Europe. At precisely that time, the death rate was one of the highest in Europe [1].

The SARS-CoV-2 virus infection is linked to high levels of proinflammatory markers; they have been analyzed in order to determine whether they could be informative with regard to the outcome of the disease. From a nephrology point of view, numerous studies were carried out in relation to COVID-19-related acute kidney injury (AKI); indeed, around 1 in 10 of hospitalized patients with COVID-19 experience AKI [2]. In addition, it was shown that AKI was a serious risk factor for the outcome; however, there is a wide variation in the incidence and the outcomes being reported [2,3].

For instance, Hirsh et al. reported a high incidence of AKI among inpatients with COVID-19 in 2020 (36.6%) [4]. In patients without COVID-19 who have a variety of underlying pathologic features, e.g., infection, cardiovascular disease, systemic disease, etc., it is believed that 20% of patients who are hospitalized will deteriorate to AKI, and 10% of patients with AKI will require renal replacement therapy (RRT) [4]. Of those who were infected with COVID-19, large surveillance studies and meta-analyses have now reported an overall incidence of AKI of 28–34% in inpatients and 46–77% in those in the intensive care unit (ICU). However, the rate of more severe AKI cases requiring RRT in the ICU appears to have reduced over time: data from England and Wales show that RRT has dropped from 26% at the start of the pandemic to 14% in 2022 [5]. Nevertheless, from the different numerical outcomes, it is crucial to point out that AKI is an absolute risk factor in relation to the death rate for patients with COVID-19 [6].

Diagnosing AKI early and taking adequate preventive measures is essential for the recovery of renal function. Presently, the standard for diagnosing AKI is based on the level of serum creatinine and urine output volume, in accordance with the recommendations issued by Kidney Disease: Improving Global Outcomes (KDIGO), in 2012 [7]. However, serum creatinine increase and urine output decrease are nonspecific and may be delayed (up to 48 h), thus obscuring the early diagnosis of AKI. It is for this reason that there is a worldwide trend to devise highly informative methods to investigate AKI in order to shorten the time to diagnosis. In an observational study it was found that sNGAL (serum neutrophil gelatinase-associated lipocalin) demonstrated lower accuracy as a diagnostic parameter [8] whereas other prospective observational study of 52 patients with COVID-19 carried out by Pode Shakked et al., they found that serum creatinine and serum cystatin C (sCysC) measured at emergency department (ED) admission were highly predictive for AKI and also reliable in predicting the need for hemodialysis [9]. In a prospective observational study of 57 COVID-19 patients admitted to the ICU, Luther et al. evaluated urinary albumin and NGAL (neutrophil gelatinase-associated lipocalin), KIM-1 (kidney injury molecule-1) and TIMP-2 (tissue inhibitors of metalloproteinases-2) from serum, on admission; the increased levels predicted AKI in the majority (89%) of patients. They found that urine markers were elevated in the majority of patients, but they did not independently predict the stage as defined by KDIGO [9].

Long before the pandemic it was well known and proved over time that AKI can develop even in non-severe cases of pneumonia, because these clinical cases can be associated with an increased immune response and at the same time with lower survival [10]. For example, Akhmerov et al. pointed out that 15% of infected patients with COVID-19 could have a complicated clinical outcome from the onset of a serious form of interstitial pneumonia, which may therefore progress towards acute respiratory distress syndrome (ARDS), and death [11]. This means that analyzing the acute inflammation process could hold the key for a better prediction of severe COVID-19, which is crucial for the clinical management of the patients.

In such severe infections like COVID-19, the macrophages are activated and that can trigger a cytokine response cascade by triggering the release of tumor necrosis factor α (TNF-α), interleukin 1 (IL-1), nitric oxide (NO) and reactive oxygen species (ROS) to induce a powerful inflammatory immune response that promotes liver cells to produce serum amyloid A (SAA) [12]. SAA is a well-known acute-phase protein produced by liver cells in response to proinflammatory cytokines released during viral infection [13]. More recent studies suggest that SAA may be a very informative predictor of the severity of disease in patients with COVID-19. Furthermore, when a high level of SAA is detected, it is more likely to result in a high degree of lung parenchymal involvement [14]. The helpfulness of monitoring SAA values as a prognostic predictor in patients with COVID-19 is also supported by other findings in which high SAA levels are related to adverse outcomes [15].

In this study, we investigated the feasibility of using SAA as a biomarker of severe COVID-19 infection, and we studied whether SAA could be a predictor of disease severity. In parallel, serum IL-6 (Interleukin-6) levels were assessed. Specifically, we aimed to determine whether SAA could be a predictor of AKI and mortality rate in the analyzed cohort; also, whether there was a direct correlation with other inflammatory markers. SAA levels were examined 24–48 h after the patients’ hospitalization and then compared with disease outcome.

## 2. Results

Of 160 individuals, 75% were confirmed positive by PCR test for COVID-19 (120 patients), of whom 58.3% (70 patients) had a history of CKD. The other 40 patients (25%) who had no history of COVID-19 were used as controls and were divided into two separate groups: 20 patients (50%) had a history of CKD, while the other 20 (50%) were completely healthy.

The median age of the CKD patients that had a positive test for COVID-19 was 56.8 years, whereas for the non-CKD COVID-19 patients, it was 65.9 years. The gender ratio was not equal in the groups. For the CKD patients without COVID-19, the median age was 66.1 years; the gender ratio was 11 females (55%) to 9 males (45%). For the healthy control group, the median age was 36.8 years, and the gender ratio was equal: 10 males and 10 females. Overall, in the COVID-19 patients, the serum creatinine level on admission was 119.0 μmol/L (57.0–930.0 μmol/L) for the CKD group and 79.0 μmol/L (50.0–295.0 μmol/L) for the non-CKD group. In addition, it was elevated over the pre-COVID-19 baseline in 58.3% of cases. For the non-COVID-19 CKD patients, the median serum creatinine was 109.1 μmol/L (62.0–188.0 μmol/L); it was elevated in 60% of cases (laboratory references: females 44–80 μmol/L; males 62–106 μmol/L). None of patients in the healthy control group had increased levels of creatinine over the baseline (for women, it was 44–80 μmol/L and for men, it was 62–106 μmol/L).

There were statistically significant differences (<0.0001) between the CRP and leukocyte values in the four groups. COVID-19 patients had higher CRP and leukocytosis. The CRP scores for COVID-19 patients were 75.7 mg/L (SD = 73.7) in the CKD group and 53.9 mg/L (SD = 59.5) in the non-CKD group. In COVID-19-negative patients, the CRP was 7.0 mg/L (SD = 6.7) in the CKD group and 1.7 mg/L (SD = 1.6) in the healthy controls. When the results for leukocytes were compared, exactly the same pattern was observed, i.e., 14.8 G/L (11.2) for CKD+COVID-19 patients and 12.1 (10.5) G/L for non-COVID-19 patients. In the COVID-19-negative patients, the leukocyte count was 8.2 G/L (SD = 6.8) in the CKD group and 4.5 G/L (SD = 2.5) in the healthy controls. The upper limit for CRP was 5 mg/L and for leukocytes was 10.5 G/L (Table 1).

Sixty-one percent of COVID-19 patients had at least one underlying comorbidity, i.e., cardiovascular disease, hypertension, diabetes, obesity, with or without CKD. A comparison was made between the baseline values of the laboratory in the different groups (Table 2). There were few significant differences between the two groups at admission: i.e., D-dimers, hemoglobin, serum creatinine level, eGFR, serum urea, proteinuria and hematuria. Overall, the serum creatinine level at presentation was increased in almost 55% of the cases; i.e., the normal range for women was 44–80 μmol/L and for men was 62–106 μmol/L. The mean eGFR at admission was 80.4 (28.9) ml/min/1.73 m^2^ for the group without CKD and 47.9 (23.0) mL/min/1.73 m^2^ for the group with CKD (*p*-value < 0.001). For the other two groups, we found significant differences once again: for the CKD patients, the mean eGFR was 62.3 (22.6) mL/min/1.73 m^2^, whereas for the healthy controls, it was 111.1 (13.0) mL/min/1.73 m^2^.

Urine samples on admission showed that 49 patients had proteinuria, 81.6% of them had 1+ proteinuria and the rest had >1+ proteinuria. We also assessed hematuria: there were 29 cases on admission, of which 48.2% had only 1+ hematuria. Fewer patients had hematuria compared to proteinuria. The incidence of ureteral catheterization on admission was quite low (<6.2%) and was expected to have little impact on urinalysis.

IL-6 and SAA levels were assessed across the four groups and the results were quite similar. When the four groups were compared with each other for both biomarkers, we had significant results only when compared to the healthy controls. There were no significant differences between the other three groups (Figure 1 and Figure 2).

There were frequent occurrences of AKI during the hospital stays, i.e., it appeared in 38 patients with COVID-19 (23.7%), of whom 31 were in the CKD COVID-19 group. This means that 34.4% in this group had AKI. COVID-19 patients in the non-CKD group had only seven cases of AKI (14%). In total, within our cohort of 120 COVID-19 patients, the in-hospital mortality rate was 14.3% (23 patients). All patients who did not survive the SARS-CoV-2 infection were in the ICU and died of progressive and complicated pneumonia; none of these patients had a de novo history of any vascular incident.

An analysis was performed to evaluate any correlations between the biomarkers (IL-6 and SAA) and the frequency of AKI in each COVID-19 group. Among the biomarkers, we found that IL-6 (*p* = 0.006) and SAA (*p* = 0.027) levels were both significantly higher in those patients who experienced AKI (Figure 3 and Figure 4).

Additional correlation analyses of biomarkers in relation to laboratory results were performed (CRP, leukocytes, neutrophils), results from the CT (computer tomography) of the lungs, ventilation (yes/no) and mortality rate. The analysis was performed to see whether any of the results had correlation with the biomarkers and were associated with severe complication of the disease and/or fatal outcome (Table 3). In COVID-19 patients who had AKI, IL-6 levels were significantly higher compared with those who did not have AKI. (median values: 20.4 vs. 5.53 pg/mL), whilst for the SAA, we had the same tendency (median values: 300.0 vs. 77.40 µg/L). It is important to point out that the cutoff value for SAA was 200 µg/L: all of the patient with results that were above 200 µg/L died no matter the period (days) of their hospitalization, whilst all the patients with results under 200 µg/L survived COVID-19 infection.

Both of the biomarkers had significant results across all comparisons that were made with only one exclusion, i.e., surprisingly, CRP has no significant correlation with IL-6. It was found that IL-6 and SAA had higher levels and were correlated with the higher number of leukocytes and neutrophils (laboratory references for leukocytes—3.5 to 10.5 g/L and for neutrophils—2.0 to 7.8 g/L). Correlation was also found between the higher levels of the two biomarkers with a greater degree of lung impairment. We have observed that all patients in our prospective study cohort who had high baseline IL-6 and SAA levels on admission required subsequent ventilation. In fact, all patients who required subsequent ventilation had IL-6 levels on admission ranging between 25 and 120 pg/mL pg/mL; thus, of the 27 COVID-19 patients who had IL-6 results >25 pg/mL on admission, 19 of them later needed mechanical ventilation). We obtained the same results for patients with SAA levels between 200 and 300 µg/mL.

COVID-19 patients all received a computerized tomography scan of the chest on arrival at the hospital after the positive test for COVID-19. Each CT scan was graded according to severity score (0–5). The scores ranged between 0 and 4 points in our patients. After data analysis, we grouped patients into two categories: score 0–2 (n = 102) and score 3–4 (n = 18). All patients in the second group experienced a fatal outcome due to the advanced pathological changes of the lung parenchyma.

An analysis was performed with a logistic regression model in order to examine factors associated with COVID-19 death. We included in the model (gender, hypertension, AKI, febrile (yes), CKD, age, SAA, IL-6, CRP, diabetes and proteinuria. Logistic regression for mortality risk factors indicated that the factors that were statistically significant, in addition to hypertension, were SAA levels, febrility, presence of AKI, and proteinuria on arrival at the ED. Hypertension was a negative prognostic factor for COVID-19 infection, i.e., it enhanced the mortality rate [OR = 12.125 (95% CI: 0.924–159.034); *p* = 0.06], as well as being febrile upon admission [OR = 8.677 (95% CI: 1.622–46.418); *p* = 0.01]; having proteinuria at admission came out as a negative risk factor [OR = 1.057 (95% CI: 0.007–0.426); *p* = 0.01]. Also, AKI came out as a significant negative factor for fatal outcome [OR = 3.634 (95% CI: 0.542–24.356); *p* = 0.01]. From both the explored biomarkers, only SAA had a negative prognostic value [OR = 1.018 (95% CI: 1.008–1.028); *p* = 0.00]. The other parameters included in the model did not predict death. (Figure 5)

## 3. Discussion

Since the beginning of the COVID-19 pandemic there have been many studies that aimed to improve the prognosis for the clinical course of the disease. Several biochemical parameters were selected and assessed as possible predictors of disease severity. The elevated levels of IL-6, D-dimers, CRP, neutrophils, and a low lymphocyte count have been associated with severe forms of COVID-19 [16,17]. In this prospective study including COVID-19 patients with or without CKD at baseline, we assessed the incremental values of the blood biomarkers (IL-6 and SAA) and evaluated them at hospitalization to identify if they could predict fatal outcome and AKI. Both serum IL-6 and SAA levels were linked to mortality and AKI. A recent systematic review showed that IL-6 significantly increases the risk of COVID-19 with a greater severity of disease (adjusted OR = 1.0284; 95% CI 1.0130–1.0441; *p* = 0.0003) and likelihood of mortality (adjusted OR = 1.0076; 95% CI 1.0004–1.0148; *p* = 0.04; adjusted hazard ratio (aHR) = 1.0036; 95% CI 1.0010–1.0061; *p* = 0.006) [18]. Another systematic review of 147 studies showed that COVID-19 patients who eventually died had 42.1-fold higher mean IL-6 concentrations than patients who survived. IL-6 levels were also elevated significantly in those who were deceased (MD: 42.11; *p* < 0.001; 95% CI: 36.86,47.36) [19].

There is increased attention being paid to the clinical value of SAA as an acute inflammation marker since the COVID-19 pandemic outburst and the evidence on that matter is accumulating. There are several studies pointing out that SAA levels in cases of severe respiratory syndrome were significantly high and it was suggested that SAA may be important for monitoring respiratory diseases [20,21,22]. A meta-analysis was performed of nineteen studies that were conducted on SAA and COVID-19: the conclusion reached was that the presence of relatively high SAA concentrations is significantly associated with more severe disease, based on clinical assessment or the presence of ARDS, and an increased risk of mortality in patients with COVID-19 [23]. In one particular study, Pieri M et al., showed that SAA levels in the serum can be very informative for COVID-19 patients admitted to hospital and that SAA might be a good prognostic marker in COVID-19 disease alone and/or in combination with other inflammatory biomarkers such as D-dimer, hsCRP, IL-6 and procalcitonin and can help distinguish the critically ill patient [15].

When we discuss the COVID-19 infection, we have to point out that the hyperinflammatory syndrome is a complex interaction between infection and autoimmunity and there is always an elevation of SAA. Recent studies also proved that there is a strong correlation between severity of the inflammation and the outcome [24]. Other studies proved that over 80% of the COVID-19 patients had elevated SAA levels from the serum [24,25]. Of course, these results were compared with healthy control groups, as we did in our study, and a significant elevation of SAA levels in COVID-19 patients was proven [18,26] and again these results showed a strong correlation between the severity and the outcome for the patients.

In our cohort, we had very strong predictive results for the outcome of the disease. All the patients who had results for SAA between 200 and 300 µg/L had died (normal result is up to 5 µg/L). The rest of the patients who had levels under 200 µg/L survived the infection. This is very important result, because, first, all the blood samples of the patients were taken in the first 24 h of their hospitalization due to COVID-19 infection and, second, a result above 200 µg/L is a sure predictor of death, regardless of the length of hospital stay. This laboratory cutoff applies inversely to patients who have survived the infection. This proves once more that SAA has very important clinical value, because the results can be an early warning of the poor prognosis and the severity of COVID-19. In relation to this, one study showed that a cut-off value of SAA for disease severity was 157.9 µg/L [27] and another one pointed out that a SAA level greater than 100.02 µg/L could be a warning sign for a patient for COVID-19 progression [28]. Also, Yang et al. proved that patients who died of COVID-19 with complications had higher SAA levels than patients who died of COVID-19 without complications; *p* < 0.01 [29]. Li et al. carried out a meta-analysis of 10 studies for SAA and severe COVID-19 infection and pointed out not only the prognostic ability of SAA for the outcome, but they go even further—SAA can also predict the recovery of the patients. [30] 

In our investigation, we also looked for early biomarkers that could be predictive of COVID-19-related AKI. We observed that IL-6 and SAA levels were significantly increased in COVID-19 patients and CKD patients without COVID-19 when compared with healthy volunteers. In COVID-19 patients, those who had AKI had significantly higher IL-6 levels compared with those who did not have AKI (20.4 vs. 5.53 pg/mL), whereas we had the same trend for SAA (300.0 vs. 77.40 µg/L). Wang et al. demonstrated that IL-6 level had significant positive correlations with serum creatinine and blood urea nitrogen [31]. In patients with COVID-19, serum IL-6 levels were elevated in those with AKI [32]. Serum IL-6 levels may also predict clinical outcomes of AKI, as they are significantly reduced in those in whom AKI was removed after effective treatment [33]. Finally, IL-6 levels of >35 pg/mL may indicate a risk of respiratory failure [34] in the context of COVID-19 infection. Here, we confirm this result, i.e., of the 27 patients with COVID-19 who had IL-6 results >25 pg/mL on admission, 19 (70.4%) of them later required mechanical ventilation [17]. For SAA, there are no data to date that can be used as a predictor of AKI—there is an analysis in relation to procalcitonin levels and acute forms of amyloidosis [35], both of which had AKI as a complication.

We investigated whether the biomarkers were associated with chest CT severity score (CTSS). Valk et al. found that despite its low prognostic ability, CTSS was linked to ICU mortality [36]. Nokiani et al. found CTSS to be an outstanding tool for triage and prognostic evaluation in COVID-19 patients aged 65 years and older [37]. In our study, we also found that COVID-19 patients with a CTSS score of between 3–4 at admission had a death rate of 88.9%, compared with only 6.8% in those with a CTSS score of 0–2 (*p*-value < 0.0001). In addition, we observed that both biomarkers (IL-6 and SAA) were associated significantly with CTSS score. In the literature, such correlations were found in only one previous study by our team [37]. Considering all our results, we therefore found that the patients who had higher levels of SAA and IL-6 on arrival and had a chest CTSS severity score >3 had a much greater risk of requiring mechanical ventilation. Sadly, all ventilated patients in our cohort had a lethal outcome.

The analysis of the risk factors performed using the odds ratio method pointed out few factors as such that can predict fatal outcome. In our cohort, having a febrile patient was found to be a negative prognostic factor [OR = 8.677 (95% CI: 1.622–46.418); *p* = 0.01], which is significant for the fatal outcome. The second significant result was proteinuria [OR = 1.057 (95% CI: 0.007–0.426); *p* = 0.01]—many reports indicate that proteinuria can be detected in AKI associated with COVID-19 (CoV-AKI) despite CoV-AKI being largely described as a form of acute tubular injury [38]. Although new-onset proteinuria has been described in some case reports in the course of COVID-19 [39], in our cohort, we had patients presenting with pre-existing CKD (90 patients), and because the patients were examined only at admission, the informative value of the result cannot be determined because we cannot avoid incorrect assertions regarding the precise onset of proteinuria. The second negative prognostic risk factor was patients having hypertension [OR = 12.125 (95% CI: 0.924–159.034); *p* = 0.06]. Although, up until this moment, there is no evidence for the precise mechanism by which hypertension predisposes to unfavorable outcomes in patients infected with COVID-19, in a few studies, it was pointed out that having good control of the blood pressure could have a favorable impact for the outcome of the disease [38]. We found that AKI is a risk factor for COVID-19-associated death [OR = 3.634 (95% CI: 0.542–24.356); *p* = 0.01]. Many papers demonstrated otherwise. A more recent study from the United States (US) indicated that of a total of 306 061 patients with COVID-19, 126,478 (41.0%) had AKI. These had a higher mortality incidence compared with the rest who did not have AKI. The severity of AKI was also associated with mortality [7]. Upper respiratory symptoms, fever, and changes in taste/smell remain the most common presenting symptoms [40]. We have also pointed out that AKI is a risk for higher mortality [5]. The last risk factor, but not the least: serum levels of SAA [OR = 1.018 (95% CI: 1.008–1.028); *p* = 0.00], has already been discussed.

This study was performed mainly with Gamma and Delta variants which at that time had become the dominant strains in many countries around the world [41]. Many analyses were performed of these variants of COVID-19, with the main goal being to analyze their impact and to better understand their pathological mechanisms. All the studies showed that patients with chronic diseases (lung, heart, kidney, etc.) had a higher mortality rate and were in greater risk of having complications like AKI and cardiac injury [42]. These two variants are pretty close in many aspects and it was a year later before Omicron was found and it was shown that the virus had lost some hallmarks such as endothelium and being more restricted with respect to cell tropism. [43]. All these studies provided similar results, i.e., a strong association between IL-6 and SAA [18] which is related proportionally to their reliability to predict the disease severity and the recovery of the patients [15,44].

Our study has some restrictions: we had 120 patients with COVID-19 which is not a high number. However, it should be noted that they were very well phenotyped. Moreover, the biomarkers were evaluated at ED admission but not afterwards, which would be interesting for a further study of the kinetics.

In the near future, we need larger-scale, multicenter studies to validate and extend the current findings. There should be collaboration between multiple healthcare centers to ensure diverse patient populations, varied demographics, and clinical characteristics. It is very important to explore how IL-6 and SAA may vary in different groups with different management strategies. The different strategies of treatment merit further investigation into the association between SAA, IL-6, and treatment response. By obtaining reliable results, it may lead to the standardization of measurement and reporting of SAA and IL-6 levels in patients with COVID-19 and CKD. Even more, the targeting of SAA is considered an early immunological intervention in most recent studies and by using a strategy to block SAA, we might inhibit the multiple cytokines involved in the inflammatory cycle and achieve a more effective outcome than those strategies where we block a single cytokine [45].

Our study has some limitations: we only had 120 COVID-19 patients; however, they were very well phenotyped. In addition, the biomarkers were only assessed at admission in the ED, but not subsequently; this would have been of interest in order to study their kinetics. Also, it is single centered which can affect the results; because the researchers were aware of the study conditions, their expectations or beliefs could inadvertently influence the results.

## 4. Materials and Methods

This single-center study was performed at Alexandrovka’s Hospital in Sofia (Bulgaria) and was conducted between 1 February 2021 and 31 March 2021. We included consecutive patients who were confirmed to have a COVID-19 infection in the ED and who were hospitalized after a positive PCR test for SARS-CoV-2. All of these cases were verified using reverse transcription-polymerase chain reaction from combined throat/nose samples [44]. Only patients older than 18 years and without urinary tract infection were included in this study. Of all 160 patients who were included in the study, none of them had been vaccinated against SARS-CoV-2. All patients in our cohort were of Caucasian race.

Of the 160 patients included in our investigation, 120 had a positive test for COVID-19. Of this group, 70 had a history of chronic kidney disease and none was on hemodialysis. Estimation of eGFR was performed by using (CKD)-EPI 2021 formula. The remaining 50 patients had no history of kidney disease and had normal serum creatinine levels. If AKI was present, it was categorized according to the KDIGO criteria. The rest of the patients in the cohort (40 in total) were also separated into two groups: 20 patients with well-characterized CKD being followed in our outpatient clinic and 20 patients who were healthy controls, and who had no history of comorbidities.

In the study, we had very strict inclusion/exclusion criteria for the patients. If a patient was selected for the CKD groups, a medical record should have been provided with a statute of limitations of at least 6 months documenting CKD. At the same time, patients who were selected for non-CKD group required a medical history from the past 6 months where data were presented for the levels of serum creatinine and also an ultrasound check was performed. A potential uroinfection was an exclusion criterion.

On admission to the hospital, all COVID-19 patients had laboratory tests that included complete blood cell counts, CRP, total serum proteins, serum albumin, D-dimers, fibrinogen, serum creatinine (eGFR CKD-EPI 2021), serum urea and a urine sample for sediment (cells), and proteinuria assessment. Two additional biomarkers were assessed, i.e., serum IL-6 and serum amyloid A (SAA) with ELISA kits obtained from BioVendor R and D. Biomarker samples (IL-6 and SAA) were stored frozen at −80°C and were thawed and then assessed the same day by the same technician.

-Serum IL-6, Catalogue number RD194015200R, produced by: BioVendor–Brno, Czech Republic. (Date of the analysis-9 September 2022);-Serum SAA—serum amyloid A (SAA) was measured on MAGLUMI 2000 (Snibe) by quantitative chemiluminescence immunoassay (SAA REF 130216005M, Lot 177230111). Normal range of SAA was up to 5 µg/dL. (Date of the analysis, 15 September 2022);

On admission to the ED, all patients with COVID-19 underwent chest CT to evaluate disease severity. We used native CT-scan. Each patient was graded in the range 0–5: score 0 (0% or none), score 1 (1–5% or minimal), score 2 (6–25% or mild), score 3 (26–49% or moderate), score 4 (50–74% or severe), and score 5 (75% or extensive) [46].

On admission to the ICU, patients were informed of the protocol by one of us (R.F.) and, after giving their informed agreement and providing signatures, blood and urine samples were collected and sent to the laboratory. The same information was provided for the non-CKD groups; both healthy volunteers and CKD controls were PCR-tested for SARS-CoV-2, which had to be negative.

The study was carried out according to the guidelines of the Declaration of Helsinki and was approved by the KENIMUS Ethics Committee at the Medical University of Sofia, Bulgaria, with protocol No. 12/31.05.2022.

### 4.1. Statistical Analysis

Categorical parameters are described by absolute and relative (percentage) frequencies. Continuous parameters are described by arithmetic means and standard deviations (SD), and median, minimum, and maximum values. The distribution of the continuous parameters was calculated for normality using the Shapiro–Wilk test.

### 4.2. Methods for Testing the Post hoc Hypotheses

To compare the continuous parameters in two related (paired) groups, the Wilcoxon two sample test was applied, and the approximation of the Student’s t-statistic (t-approximation) with a continuity correction of 0.5 was used to determine statistical significance. In addition, the sign test was used where appropriate.

Comparisons between independent (unrelated) groups (CKD vs. no CKD) were made after adjustment for baseline differences.

Due to heterogeneity and non-normality of data distribution, non-parametric ANOVA (Kruskal–Wallis) was used to compare more than two groups.

Linear association between continuous normally distributed variables was estimated by Pearson’s correlation coefficient. For continuous non-normal parameters, Spearman’s coefficient was used. Bi-serial coefficient was used to explore the relation between non-metric variables.

### 4.3. Method Used for Data Modelling

Binary logistic regression was used to model the relationship between the output (COVID-19-related death) as a dependent variable and the main parameters. A model is presented for death. In addition, the odds ratios and forest plots are shown. For decision making, a significance level of 5% was used. The SAS^®^ package version 9.4 (SAS Institute Inc., SAS 9.4 Help and Documentation, Cary, NC, USA: SAS Institute Inc., 2015–2022) was used for the calculations and the graphical presentations.

## 5. Conclusions

Biomarkers of acute inflammation like SAA and IL-6 can have a high informative value in the ongoing COVID-19 infection. They both have proved that they have correlation with other laboratory biomarkers and with lung imaging results. SAA has also proven to be a strong independent prognostic biomarker for the disease course, for prognosis of complications like AKI and also, for the outcome of the disease when it is measured in the first 24–48 h after the hospitalization of the patients.

## Figures and Tables

**Figure 1 ijms-25-00311-f001:**
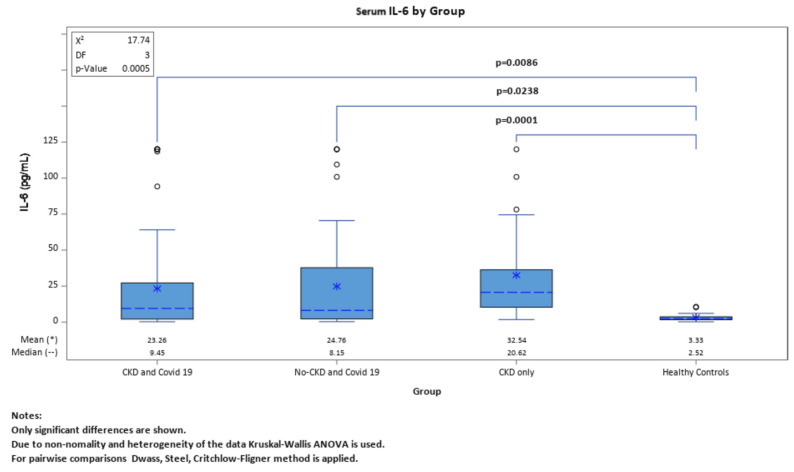
Comparing levels of IL-6 between the four groups.

**Figure 2 ijms-25-00311-f002:**
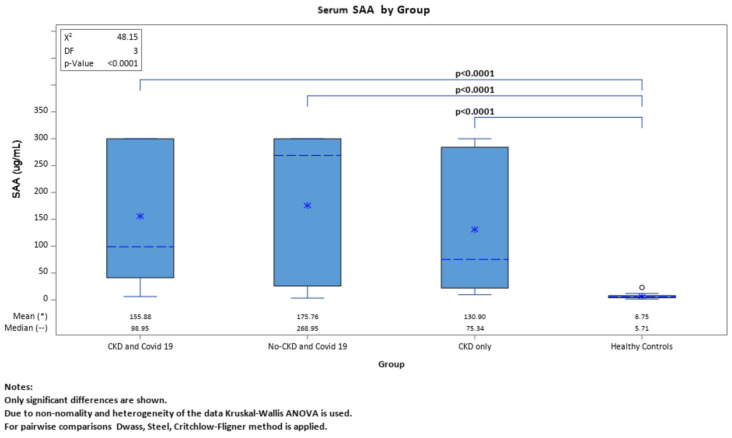
Comparing levels of SAA between the four groups.

**Figure 3 ijms-25-00311-f003:**
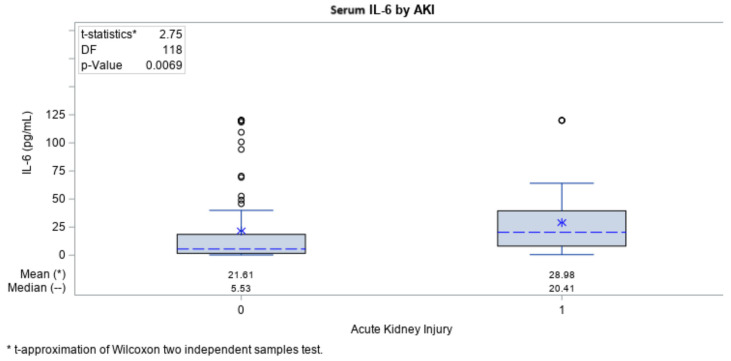
Levels of IL-6 in patients with or without AKI. Appendix: 0—no AKI; 1—documented AKI.

**Figure 4 ijms-25-00311-f004:**
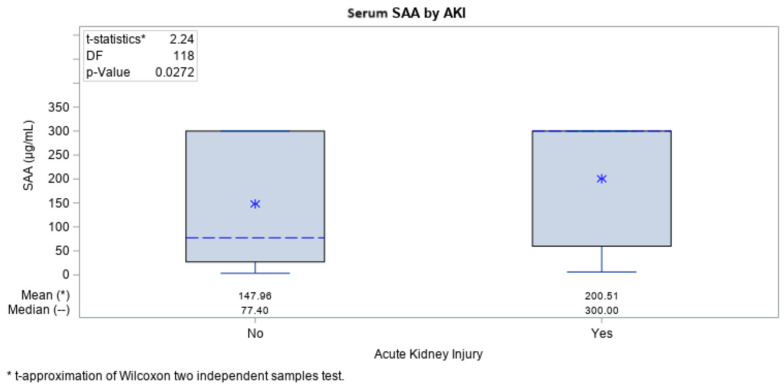
Levels of SAA in patients with or without AKI. Appendix: 0—no AKI; 1—documented AKI.

**Figure 5 ijms-25-00311-f005:**
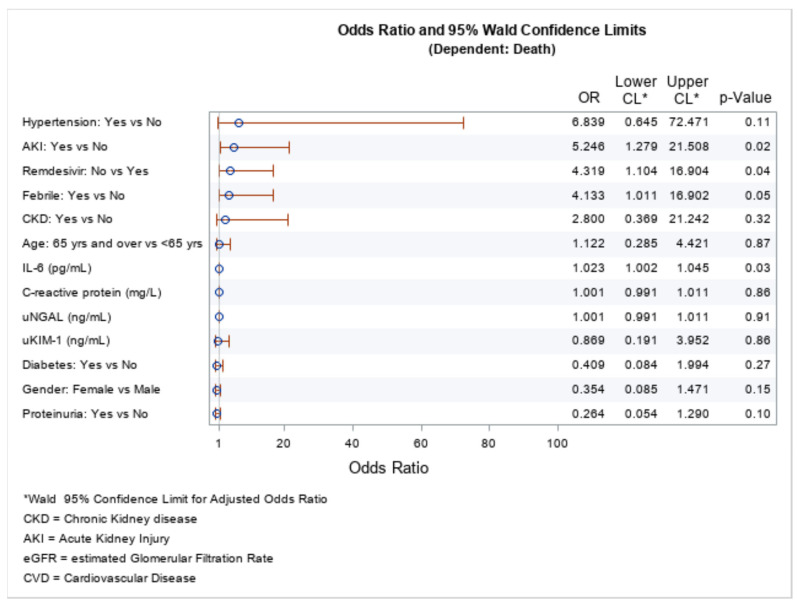
Independent risk factors for death.

**Table 1 ijms-25-00311-t001:** Description of the population.

Group	Patients	Sex	Age Mean (SD)	Race (n%)	Creatinine Level (mmol/L) Mean SD	eGFR (CKD-EPI 2021) Mean (SD)	CRP (mg/dL) Mean (SD)	Leu (g/L) Mean (SD)
**CKD and COVID-19**	70	Female: 39 (55.7%)	56.8 (16.0)	Caucasians 100%	171.4 (156.4)	47.9 (23.0)	75.7 (73.7)	14.8 (11.2)
		Male: 31 (44.3%)						
**Non-CKD and COVID-19**	50	Female: 21 (42.0%)	65.9 (12.6)	Caucasians 100%	120.8 (176.9)	80.4 (28.9)	53.9 (59.5)	12.1 (10.5)
		Male: 29 (58.0%)						
**CKD and no COVID-19**	20	Female: 11 (55%)	66.1 (11.8)	Caucasians 100%	109.1 (31.8)	62.3 (22.6)	7.0 (6.7)	8.2 (6.8)
		Male: 9 (45%)						
**Healthy controls**	20	Female: 10 (50%)	36.8 (7.8)	Caucasians 100%	68.8 (15.6)	111.1 (13.0)	1.7 (1.6)	4.5 (2.5)
		Male: 10 (50%)						
***p*-values**		0.49	<0.0001	NA	0.0229	<0.0001	<0.0001	<0.0001

Abbreviations: CKD—chronic kidney disease; CRP—C-reactive protein; eGFR—estimated glomerular filtration rate; SD—standard deviation; Leu—leukocytes.

**Table 2 ijms-25-00311-t002:** Baseline comparison.

Parameter	Statistical Analysis	CKD Patients + COVID-19	Non-CKD Patients + COVID-19	CKD Patients without COVID-19	Healthy Controls	*p*-Value
Number of patients	n	70	50	20	20	
Creatinine (mcmol/L)	Median (Ranges)	119 (57.0–930.0)	79 (50.0–1295.0)	139.1 (86.0–206.0)	60 (49.0–83.0)	<0.0001
eGFR (ml/min)	Mean (SD)	47.9 (23.0)	80.4 (28.9)	62.3 (22.6)	111.1 (13.0)	<0.0001
Urea (mmol/L)	Median (Ranges)	9.2 (2.7–75.2)	5.5 (3.0–82.3)	8.0 (5.0–23.0)	3.0 (2.8–5.0)	<0.0001
D-Dimer (mg/L FEU)	Median (Ranges)	0.9 (0.3–10.7)	0.5 (0.3–8.1)	0.7 (0.6–4.7)	0.2 (0.1–0.4)	<0.0001
Hemoglobin (g/dL)	Mean (SD)	133.7 (19.61)	142.5 (15.99)	110.2 (20.5)	156.2 (25.2)	0.0128
Leukocytes (10^9^ cpl)	Mean (SD)	14.8 (11.2)	12.1 (10.5)	8.2 (6.8)	4.5 (2.5)	<0.0001
CRP (mg/L)	Median (Ranges)	50.8 (0.5–320.6)	29.8 (0.1–217.6)	17.0 (1.1–30.0)	1.5 (0.1–5.0)	0.0003
		**Urine measurements**				
**Parameter**	**Statistical Analysis**	**CKD Patients**	**Non-CKD Patients**	**CKD Patients without COVID-19**	**Healthy Controls**	***p*-Value**
		Proteinuria (n%)				
1+	n(%)	33(47.1%)	2(4.0%)	5(25%)	0(0%)	
2+	n(%)	6(8.6%)	3(6.0%)	2(10%)	0(0%)	
3+	n(%)	5(7.1%)	1(2.0%)	3(15%)	0(0%)	
Neg.	n(%)	26(37.1%)	44(88.0%)	10(50%)	20(100%)	<0.0001
		Hematuria (n%)				
1+	n(%)	12(17.1%)	0(0.0%)	2(10%)	0(0%)	
2+	n(%)	4(5.7%)	1(2.0%)	2(10%)	0(0%)	
3+	n(%)	3(4.3%)	2(4.0%)	3(15%)	0(0%)	
Neg.	n(%)	51(72.9%)	47(94.0%)	13(65%)	20(100%)	0.0056

Abbreviations: CKD, chronic kidney disease; eGFR, estimated glomerular filtration rate; CRP, C-reactive protein; Neg, negative; SD, standard deviation.

**Table 3 ijms-25-00311-t003:** Correlations between biomarkers and inflammatory laboratory parameters.

Biomarker	IL-6	SAA
**Leukocytes** ***p***-**value**	0.18728 0.04	0.12048 0.01
**Neutrophils** ** *p* ** **-value**	0.35128 <0.0001	0.28128 <0.0001
**CRP** ** *p* ** **-value**	0.01911 0.83	0.24486 0.04
**CT ** ** *p* ** **-value**	0.32857 0.0003	0.35843 0.0002
**Ventilation** ** *p* ** **-value**	0.27175 0.002	0.27195 0.005
**Death** ** *p* ** **-value**	0.27175 0.002	0.32457 0.001

Abbreviations: IL-6, Interleukin 6; SAA, serum amyloid A; CRP, C-reactive protein; CT, lung computer tomography.

## Data Availability

All of the data are available upon request from the corresponding author.

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
