# Peer review of "IL-6 and SAA—Strong Predictors for the Outcome in COVID-19 CKD Patients"

_ijms, 2023, doi:10.3390/ijms25010311_

Round 1
Reviewer 1 Report
Comments and Suggestions for Authors
Authors show an interesting study analyzing predictors of worse outcome in COVID-19 patients with CKD. IL-6 and SAA are known markers of inflammatory reactions, and analyzing their serum level in CKD patients (who already have higher inflammatory markers level) in COVID-19 setting brings some interesting data about prognosis of final outcome.
However:
1) please explain abbreviations when used for the first time, even in the abstract, i.ex. IL-6 and SAA (L16), NGAL, KIM-1 (line 70), TIMP-2 (line 71), and do not repeat full names of abbreviations later (AKI line 161 or CRP line 352, CT lines 363 and 364),
2) please be more specific and do not explain twice what you've found, i.ex. in lines 21-23 'biomarkers were statistically different' and then 'were significantly increased',
3) please be more specific and add in line 28 why IL-6 and SAA are useful markers for COVID19 patients, markers of what? and for all COVID19 patients? in the title you've focused on CKD,
4) lines 32-34: I suggest to maybe say that almost 7 million people died till July 2023 (or 07.07.2023), not untill this moment; and please change the website link into reference;
5) please write appropriately that you say about SARS-CoV2 infection, not COVID-19 virus infection, COVID-19 is not a virus (line 38); similar, please write COVID-19 in one way, not COVID19;
6) I suggest to shorten data and linked them, they look llike seperated and without reference (lines 68-69),
7) please add that sNGAL, KIM-1, TIMP-2, etc. level (was it always in the serum?) predicts AKI (lines 68-73),
8) is MOF always impairing 'normal kidney function'? (lines 80-81), I suggest to remove this sentence, MOF from definition is an organ damage,
9) if you've analyzed in control group 20 CKD patients (line 106), why gender ratio was 11 F to 10 M (line 111),
10) the data you're showing are presented in different form, some are median (liek creatinine), some have SD (like CRP), please try to unify them, and in line 118 there are some brackets left;
11) please try to not write that 'no patients in a healthy control group had creatinine levels over the baseline', rather say whar was the level;
12) what are the 'leukocyte results'? (line 121), leukopenia or leucocytosis?
13) Table 1: data among CKD+COVID-19 patients seem to be misplaced in the 'Sex' column, and please add 'LEU' abbreviation explanation;
14) please write p-value result as just p < 0.001 (line 141),
15) Table 2: can you please show direct values of p, not just 0.0000,
16) Figures 1-4, can you please add in the caption 'serum',
17) please use appropriately citing Authors data, rather only surnames are used, i.ex. Nokiani et al. (line 383), and check references numbers, this article is not under 30 positions (like in the text line 284).
Comments on the Quality of English Language
Some minor corrections are needed, i.ex. 'hemodialisys' (line 65), 'urea proteinuria' instead of 'urea' and 'proteinuria', 'L-60' (line 270), 'AKI is a risk for... (line 315).
Reviewer 2 Report
Comments and Suggestions for Authors
Upon reviewing the paper "IL-6 and SAA – strong predictors for the outcome in COVID-19 CKD patients", the following drawbacks were identified:
1. The study's sample size of 120 COVID-19 patients, while adequate for initial analysis, is relatively small. This limitation affects the generalizability of the findings to the broader population of COVID-19 patients with chronic kidney disease (CKD).
2. Some aspects of the methodology could be elaborated upon. For instance, the paper could provide more details on the criteria for selecting patients and controls, as well as the specifics of the statistical analyses used.
3. The discussion section could benefit from a deeper analysis of the results, particularly in the context of existing literature. Comparing and contrasting the findings with previous studies would provide a more comprehensive understanding of their implications.
4. While the study acknowledges its single-center nature, it does not extensively discuss the potential biases that may arise from this setup. A more detailed examination of how the study's design might influence the results would strengthen its conclusions.
5. The paper could more explicitly outline potential directions for future research, especially in terms of larger-scale studies or different patient demographics, to validate and expand upon the findings presented.
6. Line 33 - maybe it is better to change the website to citation of the website?
Overall, while the study offers valuable insights into IL-6 and SAA as predictors in COVID-19 patients with CKD, addressing these drawbacks could enhance its robustness and applicability.
Author Response
Review Report – Reviewer #2
We are very grateful for the positive review report given by Reviewer #2 and the chance for us to update and improve our paper. We will answer/improve our paper point by point.
Comments and Suggestions for Authors
Upon reviewing the paper "IL-6 and SAA – strong predictors for the outcome in COVID-19 CKD patients", the following drawbacks were identified:
- The study's sample size of 120 COVID-19 patients, while adequate for initial analysis, is relatively small. This limitation affects the generalizability of the findings to the broader population of COVID-19 patients with chronic kidney disease (CKD).
Thank you for the comment done by the Reviewer #2. We would like to emphasize that we have cohort of 120 patients, because we had very strict criteria for inclusion in the study. At the moment when it was performed the cases of COVID-19 were really high and we could have easily have gathered a greater number of patients, but we needed patients that met certain inclusion criteria. In the lower comment (Comment #2) we have provided our inclusion criteria.
We do realize that 120 patients are a small size cohort, but at the same time we have a very homogeneous group of patients, which makes the conclusions and results of our study very strong and without external modifying factors.
- Some aspects of the methodology could be elaborated upon. For instance, the paper could provide more details on the criteria for selecting patients and controls, as well as the specifics of the statistical analyses used.
In this comment is pointed out that we should provide more details on the criteria for selection our patients, which is indeed something we have missed when the paper was written. We would like to share our inclusion criteria for the patients:
- If the patients had CKD they should have had a medical document proving that with a limitation period of at least 6 months. If such document was not provided the patient was newly discovered was not included in the study.
- Patients who were added to the non-CKD group should have provided medical history form the past 6 months where we had data for the levels of serum creatinine, ultrasound and data about the medications that they have used for the same period. If such documents were not provided the patient was not included.
- All the patients were tested for potential Uroinfection and if the sediment cast and microbiology proved such thing the patient was excluded from the study.
- Also, if the PCR-result from nose and throat came out as “dubious result” and was not taken second time – the patient was not included.
- We must also point out that all of the patients were included in the first 24 hours (since their hospitalization) in the study. Most of them were scared and afraid and not every person had the mentality to participate and to sign informed consent, because they were scared on the first place for their life. This was also a factor that we had to cope with.
In relation to the statistical analysis – it is improved in the paper.
Correction:
In the study we had very strict inclusion/exclusion criteria for the patients. If a patient was selected fort the CKD-groups a medical record should have been provided a medical record with a statute of limitations of at least 6 months documenting CKD. At the same time patients who were selected for non-CKD group should have had medical history form the past 6 months where data was presented for the levels of se-rum creatinine and also an ultrasound check was performed. A potential uroinfection was exclusion criteria.
Statistics:
To compare the continuous parameters in two related (paired) groups, the Wil-coxon two sample test was applied, and the approximation of the Student’s t-statistic (t-approximation) with a continuity correction of 0.5 was used to determine statistical significance. In addition, the sign test was used where appropriate.
Comparisons between independent (unrelated) groups (CKD vs. no CKD) were made after adjustment for baseline differences.
Due to heterogeneity and non-normality of data distribution non-parametric ANOVA (Kruskal-Wallis) was used to compare more than two groups.
Linear association between continuous normally distributed variables was esti-mated by Pearson’s correlation coefficient. For continuous non-normal parameters Spearman’s coefficient was used. Bi-serial coefficient was used to explore the relation between non-metric variables.
- The discussion section could benefit from a deeper analysis of the results, particularly in the context of existing literature. Comparing and contrasting the findings with previous studies would provide a more comprehensive understanding of their implications.
Thank you for the suggestion. We have added more results from the existing literature in the discussion part which are compared to our work.
Corrected Discussion part:
When we are discussing the COVID-19 infection we have to point out that the hyperinflammatory syndrome is a complex interaction between infection and auto-immunity and there is always an elevation of SAA. The resent studies also proved that there is a strong correlation with severity of the inflammation and the outcome [42]. Other studies proved that over 80% of the COVID-19 patients had elevated SAA level from the serum [43, 44]. Of course, these results are compared with healthy control groups, as we have done in our study and significant elevation of SAA levels in COVID-19 patients proved [45, 46] and again these results had strong relation to the severity and the outcome for the patients.
In our cohort we had very strong predictive results for the outcome of the disease. All the patients who had results for SAA between 200-300 µg/L had died (normal result is up to 5 µg/L). The rest of the patients who had under 200 µg/L have survived the infection. This is very important result, because firstly all the blood samples of the patients were taken in the first 24 hours of their hospitalization due to COVID-19 infection and second, a result above 200 µg/L is a sure predictor of death, regardless of the length of hospital stay. This laboratory cutoff applies inversely to patients who have survived the infection. This proves once more that SAA has very important clinical value, because the results can be an early warning of the poor prognosis and the sever-ity of COVID-19. In relation to that one study showed that a cut-off value of SAA for disease severity is 157.9 µg/L [47] and another one pointed out that a SAA level greater than 100.02 µg/L can be a warning sign for a patient for the COVID-19 progression [48]. Also, Yang et al. proved that patients died of COVID-19 complications had higher SAA levels than patients died of COVID-19 without complications; p < 0.01 [49]. Li et al. have done a meta-analysis of 10 studies for SAA and severe COVID-19 infection and have pointed out not only the prognostic ability of SAA for the outcome, but they go even further – SAA can predict also the recovery of the patients. [38]
In our investigation, we also looked for early biomarkers that could be predictive of COVID-19-related AKI. We observed that IL-6 and SAA levels were significantly increased in COVID-19 patients and CKD patients without COVID-19 when compared with healthy volunteers. In COVID-19 patients, those who had AKI had a significantly higher IL-6 levels compared with those who did not have AKI (20.4 vs. 5.53 pg/mL), whereas we had the same trend for SAA (300.0 vs. 77.40 µg/L). Wang et al. demonstrated that IL-6 level had significant positive correlations with serum creatinine and blood urea nitrogen [25]. In patients with COVID-19, serum IL-6 levels were elevated in those with AKI [26]. Serum IL-6 levels may also predict clinical outcomes of AKI, as they are significantly reduced in those in whom AKI was removed after effective treatment [27]. Finally, IL-6 levels of >35 pg/ml may indicate a risk of respiratory fail-ure [28] in the context of COVID-19 infection. Here, we confirm this result, i.e., of the 27 patients with COVID-19 who had IL-6 results >25 pg/mL on admission, 19 (70.4%) of them later required mechanical ventilation For SAA, there are no data to date that can be used as a predictor of AKI - there is an analysis in relation to procalcitonin levels and acute forms of amyloidosis [35], both of which had AKI as a complication.
- While the study acknowledges its single-center nature, it does not extensively discuss the potential biases that may arise from this setup. A more detailed examination of how the study's design might influence the results would strengthen its conclusions.
Thank you for the suggestion. It is updated in the discussion section.
Acknowledging the limitations of a study, such as its single-center nature and the relatively small sample size of 120 COVID-19 patients, is an essential aspect of scientific research.
Our study has some limitations: we only had 120 COVID-19 patients; however, they were very well phenotyped. In addition, the biomarkers were only assessed at admission in the ED, but not subsequently; this would have been of interested in order to study their kinetics. Also, it is single centered which can affect the results, because the researchers were aware of the study conditions, their expectations or beliefs could inadvertently influence the results.
- The paper could more explicitly outline potential directions for future research, especially in terms of larger-scale studies or different patient demographics, to validate and expand upon the findings presented.
Thank you for the suggestion. It is updated in the discussion section.
In the near future we need larger-scale, multicenter studies to validate and extend the current findings. There should be collaboration between multiple healthcare centers to ensure diverse patient populations, various demographic, and clinical characteristics. It is very important to explore how IL-6 and SAA may vary in different groups with different management strategies. The different strategies of treatment can have another investigation on the association between SAA, IL-6, and treatment response. Obtaining reliable results may lead to the standardization of measurement and reporting of SAA and IL-6 levels in patients with COVID-19 and CKD.
- Line 33 - maybe it is better to change the website to citation of the website?
Thank you for the suggestion. It is updated in the paper.
Correction:
Since the outburst of COVID-19 pandemic almost 7 million people have died up until this moment (https://covid19.who.int/?mapFilter=deaths (accessed on 13.12.2023). From a medical point of view, the situation in Bulgaria throughout the pandemic was not too much different from the situation in the most of the world, but according to official data, the percentage of people vaccinated in Bulgaria is one of the poorest in Europe. At precisely that time, the death rate was one of the highest in Eu-rope [1].
Overall, while the study offers valuable insights into IL-6 and SAA as predictors in COVID-19 patients with CKD, addressing these drawbacks could enhance its robustness and applicability.